# Dendritic mitochondria reach stable positions during circuit development

**Michelle C Faits[1,2], Chunmeng Zhang[1], Florentina Soto[1], Daniel Kerschensteiner[1,3,4,5]***

[1]Department of Ophthalmology and Visual Sciences, Washington University School of Medicine, Saint Louis, United States; [2]Graduate Program in Developmental, Regenerative and Stem Cell Biology, Washington University School of Medicine, St. Louis, United States; [3]Department of Neuroscience, Washington University School of Medicine, Saint Louis, United States; [4]Department of Biomedical Engineering, Washington University School of Medicine, Saint Louis, United States; [5]Hope Center for Neurological Disorders, Washington University School of Medicine, Saint Louis, United States

**\*For correspondence:**
dkerschensteiner@wustl.edu

**Competing interests:** The authors declare that no competing interests exist.

**Abstract** Mitochondria move throughout neuronal dendrites and localize to sites of energy demand. The prevailing view of dendritic mitochondria as highly motile organelles whose distribution is continually adjusted by neuronal activity via $Ca^{2+}$-dependent arrests is based on observations in cultured neurons exposed to artificial stimuli. Here, we analyze the movements of mitochondria in ganglion cell dendrites in the intact retina. We find that whereas during development 30% of mitochondria are motile at any time, as dendrites mature, mitochondria all but stop moving and localize stably to synapses and branch points. Neither spontaneous nor sensory-evoked activity and $Ca^{2+}$ transients alter motility of dendritic mitochondria; and pathological hyperactivity in a mouse model of retinal degeneration elevates rather than reduces motility. Thus, our findings indicate that dendritic mitochondria reach stable positions during a critical developmental period of high motility, and challenge current views about the role of activity in regulating mitochondrial transport in dendrites.

## Introduction

Mitochondria provide energy in the form of ATP and phosphocreatine, and participate in $Ca^{2+}$ signaling. In neurons, mitochondrial biosynthesis occurs in the soma, but sites of energy use and $Ca^{2+}$ influx are dispersed across axonal and dendritic arbors (*Davis and Clayton, 1996*). To meet these distributed demands, neuronal mitochondria are transported throughout axons and dendrites along microtubule tracks (*Ehlers, 2013*; *Lin and Sheng, 2015*). A number of recent studies have explored the dynamics and function of axonal mitochondria, including detailed analyses of mitochondrial movements in peripheral and central axons in vivo (*Breckwoldt et al., 2014*; *Misgeld et al., 2007*; *Plucinska et al., 2012*; *Takihara et al., 2015*). By comparison, dendritic mitochondria are less explored and their dynamics have not been examined in intact neural circuits.

Most of the energy in dendrites is consumed at synapses (*Attwell and Laughlin, 2001*; *Howarth et al., 2012*), which are also a primary site of $Ca^{2+}$ influx (*Augustine et al., 2003*; *Grienberger and Konnerth, 2012*). Dendritic mitochondria have been shown to localize to synapses in several systems and appear to contribute to their formation and plasticity (*Chang et al., 2006*; *Ishihara et al., 2009*; *Li et al., 2010*; *Li et al., 2004*). Increased ER complexity and Golgi outposts at dendritic branch points support protein and lipid biosynthesis, and secretory trafficking (*Cui-Wang et al., 2012*; *Ehlers, 2013*; *Horton et al., 2005*; *Ye et al., 2007*). In addition, branch points

**eLife digest** Inside the cells of animals and plants, compartments called mitochondria play several important roles including supplying chemical energy for cellular processes. The mitochondria inside nerve cells are produced in the main body of each cell, and must travel down a long nerve fiber called the axon and the branch-like extensions called dendrites to reach the sites where they are most needed. As the nerve cells form, dendritic branches grow and retract as the connections between different nerve cells – known as synapses – form and disappear. Later on, the dendrites and synapses become more stable, but it is not clear if the amount that the mitochondria move also changes.

Faits et al. used microscopy to study the movement of mitochondria in the developing dendrites of ganglion cells in the eyes of mice. The experiments show that early on in the development of nerve cells, the mitochondria are very mobile. However, as the synapses become more stable later on, the mitochondria become almost motionless. The movement of another type of cell compartment to the dendrites is unaffected, which suggests that this decline in movement is specific to mitochondria.

Next, Faits et al. studied mutant mice that suffer from degeneration of part of the eye called the retina. These mice have ganglion cells that display higher levels of spontaneous activity than normal and their synapses continue to form and disappear later in development. The experiments show that the mitochondria in the ganglion cells remain mobile in the adult mutant mice. Faits et al.'s findings challenge the prevailing views of mitochondria in dendrites, and suggest that mitochondria reach stable positions during a critical period in the development of the retina.

Further studies should reveal how the decline in the movement of mitochondria is regulated, which may help us to understand how differences in the movement of mitochondria can lead to the degeneration of nerve cells in some human diseases, such as dominant optic atrophy.

are hotspots for $Ca^{2+}$ signals in some dendrites (*Fitzpatrick et al., 2009*; *Larkum et al., 2003*). Mitochondria are required for dendrites to establish normal branching patterns (*Fukumitsu et al., 2015*; *Kimura and Murakami, 2014*), but whether mitochondria localize to branch points is not clear. Moreover, when associations between mitochondria and synapses or branch points emerge during development has not been examined. Finally, the prevailing view of dendritic mitochondria as being highly motile is based on observations of cultured neurons isolated from embryonic tissue. Whether this view holds true for neurons in their native environments, and how mitochondrial motility in dendrites changes as branching and connectivity patterns stabilize during circuit maturation remains unknown.

Neuronal activity is thought to control the movements of dendritic mitochondria. In particular, increases in intracellular $Ca^{2+}$ were shown to uncouple mitochondria from motor proteins via the $Ca^{2+}$-dependent adaptor protein Miro1 (*MacAskill et al., 2009*; *Wang and Schwarz, 2009*). However, the evidence supporting a leading role for activity in dendritic mitochondrial transport was obtained from cultured neurons exposed to artificial stimuli (e.g. high external $K^+$), raising the question whether physiologically occurring activity patterns in intact circuits exert a similar influence.

Here, we analyze how development and neuronal activity shape the distribution and dynamics of mitochondria in dendrites of ganglion cells in the intact retina. We find that mitochondrial density in retinal ganglion cell (RGC) dendrites reaches near-mature levels before most synapses are formed. In the adult retina, mitochondria are enriched at synapses and branch points of RGC dendrites. Mitochondria localize to synapses early, whereas their accumulation at branch points occurs gradually during circuit development. The motility of mitochondria changes fundamentally across development. During the period of dendrite growth and synaptogenesis approximately 30% of mitochondria are in motion at any time. By contrast, we observed no movements of dendritic mitochondria in mature circuits. This drastic decline in motility is cargo-specific, as peroxisomes remain motile in mature RGC dendrites. Using simultaneous two-photon imaging of mitochondrial movements and intracellular $Ca^{2+}$, we find that although elevation of external $K^+$ reduces motility of dendritic mitochondria in the retina as it does in cultured neurons, neither spontaneous waves of activity during

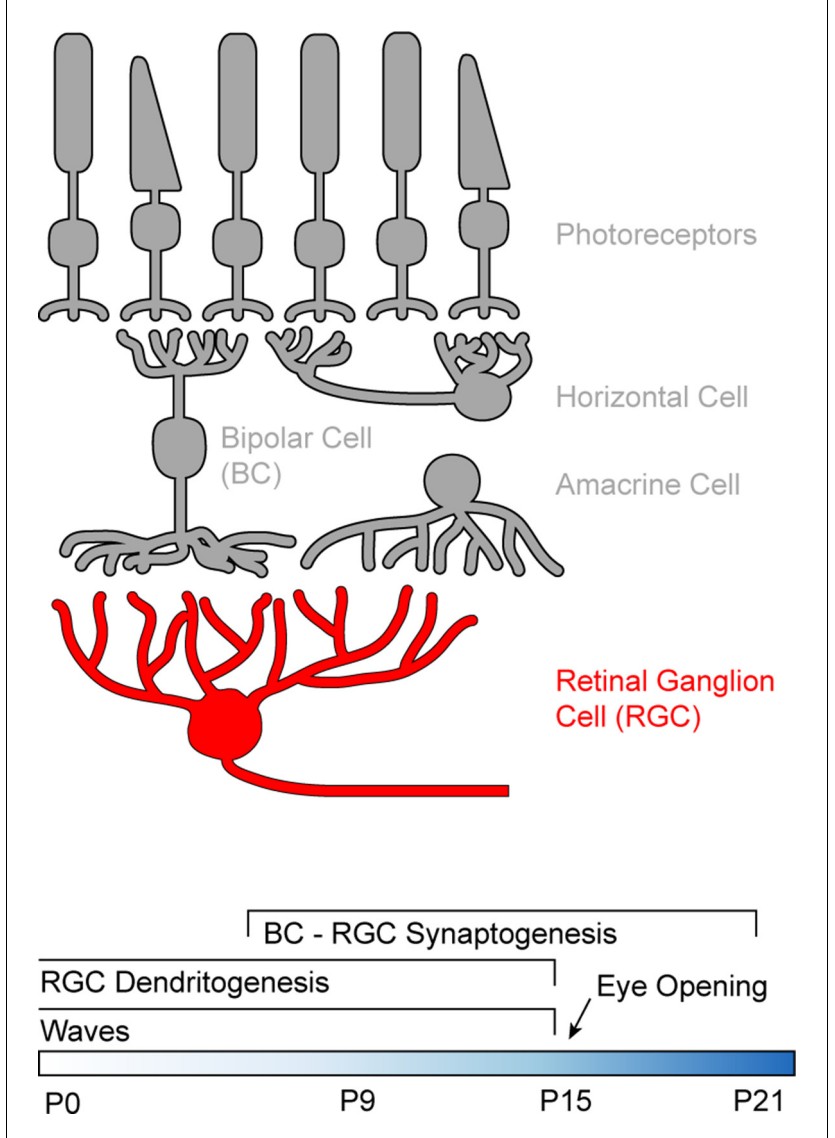

**Figure 1.** Schematic of the retinal circuitry and its development. *Top*: Illustration of the five classes of neurons in the mammalian retina. Photoreceptors - rods and cones indicated by the shape of their outer segments - translate changes in photon flux into changes in glutamate release. Horizontal cells provide feedback to photoreceptor terminals, whereas bipolar cells (BCs) relay photoreceptor signals from the outer to the inner retina where they provide excitatory input to a diverse class of inhibitory interneurons called amacrine cells and to retinal ganglion cells (RGCs), the output neurons of the eye. *Bottom*: Timeline of retinal development from birth (postnatal day 0) to circuit maturity (P21).

development, nor sensory-evoked activity at maturity alter the motility of dendritic mitochondria in RGCs. Finally, pathologically elevated RGC activity, a common feature of retinal degeneration, restores motility in mature retinas to developmental levels.

## Results

We used RGCs to study how development and neuronal activity influence the distribution and dynamics of dendritic mitochondria. In the mature retina, photoreceptors translate changes in photon flux into changes in glutamate release (*Figure 1*), and bipolar cells (BCs) relay this signal from the outer to the inner retina where they provide excitatory input to the dendrites of RGCs

(*Euler et al., 2014*; *Masland, 2001*). In mice, synapses between BCs and RGCs begin to form around postnatal day 7 (P7), and connectivity patterns reach maturity around P21 (*Fisher, 1979*; *Johnson et al., 2003*; *Morgan et al., 2008*). Dendrite development leads synaptogenesis, and arborization patterns become set by P15 (*Kim et al., 2010*; *Morgan et al., 2008*; *Soto et al., 2012*). During development, RGCs exhibit spontaneous waves of activity, which propagate through the early visual system and shape its synaptic organization (*Ackman et al., 2012*; *Kerschensteiner, 2013*; *Meister et al., 1991*). Retinal waves stop around the time of eye opening (P14-15 in mice) when sensory-evoked inputs begin to drive RGC activity (*Demas et al., 2003*).

To analyze the distribution and dynamics of mitochondria in RGC dendrites, we performed confocal and two-photon imaging in flat-mounted explants, which leave the circuitry of the retina intact (*Williams et al., 2013*). We have previously shown that BC-RGC synaptogenesis in retinal flat mounts matches in vivo rates (*Kerschensteiner et al., 2009*; *Morgan et al., 2011*; *Soto et al., 2012*). Moreover, during development, retinal circuits in this preparation generate spontaneous waves of activity with patterns similar to those observed in vivo (*Ackman et al., 2012*; *Demas et al., 2003*); and at maturity light, responses of RGCs in retinal explants closely resemble those recorded in vivo (*Farrow et al., 2013*; *Grimes et al., 2015*; *Sagdullaev and McCall, 2005*). Retinal explants thus allow us to examine the influence of development, and spontaneous and sensory-evoked physiologic activity patterns on dendritic mitochondria.

## Development of the distribution of dendritic mitochondria

To analyze the distribution of dendritic mitochondria relative to sites of high energy demand and $Ca^{2+}$ signaling, and assess its changes across development, we biolistically labeled RGCs with mitochondrially targeted yellow fluorescent protein (mtYFP), the postsynaptic density protein 95 fused to CFP (PSD95-CFP), and a red cytosolic fluorophore (tdTomato). Others and we previously showed that PSD95-CFP localizes specifically to BC synapses on RGC dendrites (i.e. excitatory synapses) (*Jakobs et al., 2008*; *Kerschensteiner et al., 2009*; *Morgan et al., 2008*). We labeled RGCs at three different ages (*Figure 2A–C*): P9 (early synaptogenesis, mid dendritogenesis, spontaneous retinal waves), P15 (mid synaptogenesis, mature dendrites, transition from retinal waves to light-evoked activity), and P21 (mature synapses, mature dendrites, light-evoked activity). We found that mitochondria enter dendrites to near-mature levels before most synapses are formed (*Figure 2D,E*). Similar to other systems, mitochondria in RGC dendrites are found closer to synapses than expected by chance (*Figure 2F*). This synaptic localization of mitochondria was apparent from the earliest time point examined (P9). In addition, at maturity, mitochondria are enriched >15-fold at branch points. Unlike their localization to synapses, the enrichment of mitochondria at branch points emerges gradually during development (*Figure 2G*). Thus, mitochondria enter RGC dendrites during early development, associate with synapses as circuits go through the trial-and-error process of establishing precise connections, and progressively accumulate at branch points even after mature arborization patterns are established.

## Cargo-specific developmental decline in mitochondrial motility

To study the dynamics of dendritic mitochondria across development, we performed time-lapse imaging experiments (0.9 frames per second or fps) on RGCs biolistically or transgenically (*Thy1-mtCFP-P*) labeled with mtYFP and mtCFP, respectively (*Misgeld et al., 2007*; *Williams et al., 2013*). Because mitochondria co-localize with synapses even at ages when synapse turnover is high (*Figure 2F*), we expected mitochondria during development to be motile, allowing them to adjust to changing patterns of connections. Indeed, similar to studies on cultured neurons, we found that at P9 approximately 30% of mitochondria moved at least once per minute (i.e. motile fraction) (*Figure 3A,B* and *Video 1*). However, subsequently, the motile fraction of mitochondria in RGC dendrites decreases steeply, falling to approximately 15% by P15, and at P21, we observed no mitochondrial movements in dendrites of 7 RGCs imaged in four retinas for 35 min. This drastic decline in motility is specific to mitochondria and does not reflect a general transition in dendritic trafficking, as SKL-GFP-labeled peroxisomes (*Monosov et al., 1996*) in RGC dendrites remain motile at P21 (*Figure 3C,D*) and beyond (*data not shown*). Interestingly, the movement patterns - including duration of runs, speed during uninterrupted motion, and duration of short pauses - of motile mitochondria at P15 were not significantly different from those observed at P9 (*Figure 3E–G*). Thus, as

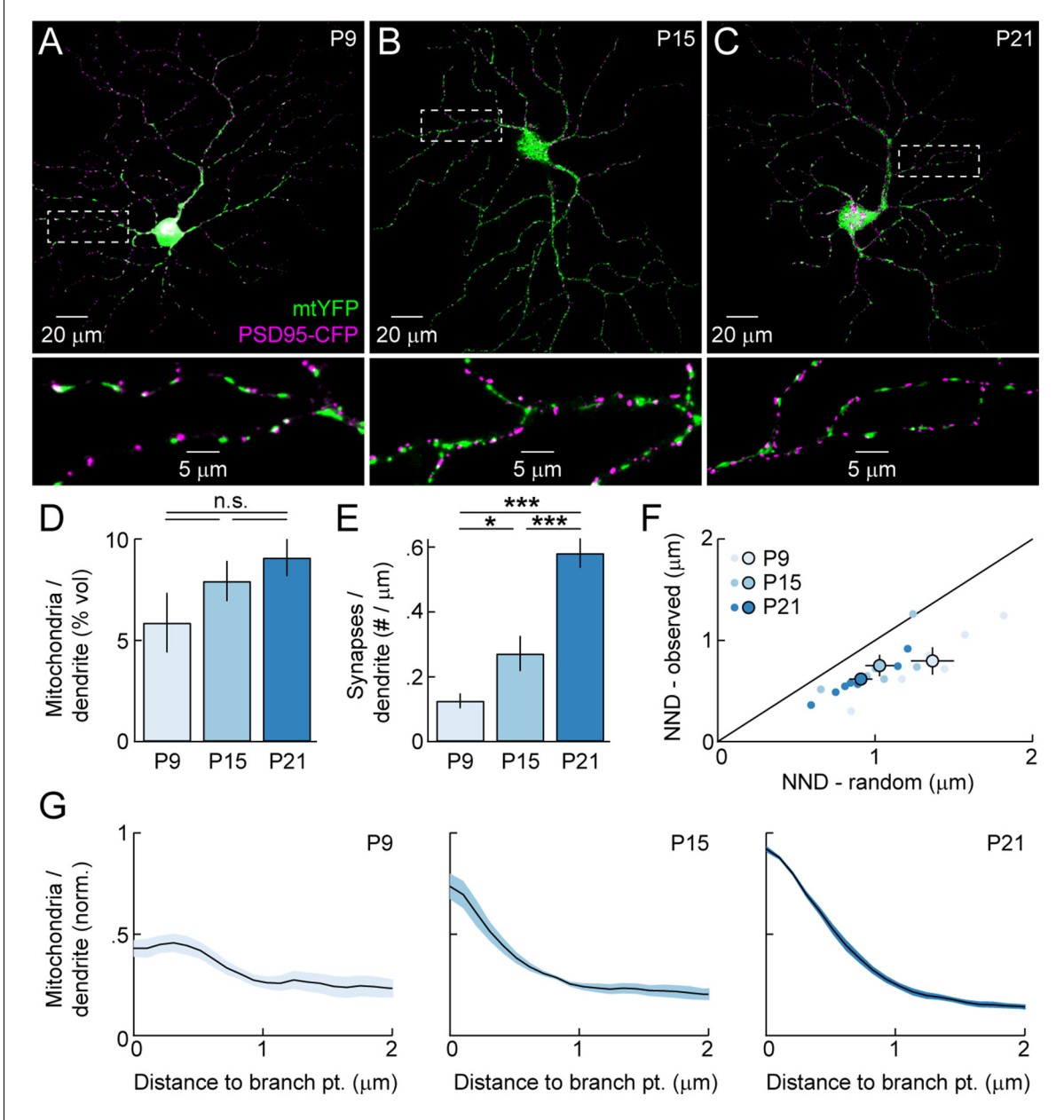

**Figure 2.** Mitochondrial distribution in RGC dendrites across development. (**A–C**) Representative RGCs expressing mtYFP and PSD95-CFP in P9 (**A**), P15 (**B**), and P21 (**C**) retinas. *Top panels* show maximum intensity projections (MIPs) through confocal image stacks and *bottom panels* show MIPs of excerpts of the same stacks at higher magnification. (**D**) Density of mitochondria in RGC dendrites expressed as a volumetric fraction (see 'Materials and methods') across development. (**E**) Density of synapses along RGC dendrites given per length of dendrite based on skeletonization of the respective arbors (see 'Materials and methods') across development. (**F**) Scatter plots comparing the average nearest neighbor distance from synapses to mitochondria (NND-observed) to the mean average NND obtained from Monte Carlo simulations in which the positions of synapses along dendrites were randomized (NND-random). *Dots* represent individual cells and *circles* (*error bars*) indicate the mean (± SEM) at the different ages examined. (**G**) Normalized mitochondrial density plotted as a function of distance from dendritic branch points at P9 (*left*), P15 (*middle*), and P21 (*right*). *Solid lines* (*shaded areas*) indicate the mean (± SEM) across a number of RGCs (P9 n = 6, P15, n = 6, P21, n = 8). mtYFP, mitochondrially targeted yellow fluorescent protein; RGC, retinal ganglion cell.

circuits mature dendritic mitochondria undergo a cargo-specific transition from a partially motile to a stationary phase, in which they preferentially localize to synapses and branch points.

# Patterned spontaneous activity and dendritic mitochondria during development

Neuronal activity has been proposed to control the motility of dendritic mitochondria. In particular, increases in intracellular $Ca^{2+}$ accompanying neuronal activation are thought to uncouple mitochondria from microtubule motors, causing acute movement arrests (*MacAskill et al., 2009*; *Wang and Schwarz, 2009*; *Yi et al., 2004*). The evidence supporting this model comes from studies of cultured neurons exposed to artificial stimuli, raising the question whether physiologically occurring activity patterns and $Ca^{2+}$ transients similarly control the movements of dendritic mitochondria in intact circuits.

The developing retina spontaneously generates propagating waves of RGC activity. In retinal waves, excitatory input (P0 – P10: cholinergic, P10 – P15: glutamatergic) elicits bursts of action potentials in RGCs (*Kerschensteiner, 2013*). As waves propagate across the retina, they synchronize the firing of neighboring RGCs. We first recorded ensembles of RGCs at P9 on multielectrode arrays (MEAs, *Figure 4A*), confirming that biolistically labeled explants generate waves of activity with similar frequency and correlation structure to those observed in unlabeled preparations and in vivo (*Figure 4C, D*) (*Ackman et al., 2012*; *Demas et al., 2003*). We then performed two-photon $Ca^{2+}$ imaging of RGCs biolistically labeled with GCaMP6s. At P9, nearby RGCs (<200 µm between cell bodies) exhibited synchronous global $Ca^{2+}$ transients across their dendritic arbors. The frequency and correlation of these transients was indistinguishable from that of spike bursts in MEA recordings (*Figure 4B–D*), indicating that global $Ca^{2+}$ transients in RGC dendrites accompany retinal waves (*Lohmann et al., 2002*). To test whether this physiologically occurring activity pattern regulates the motility of mitochondria in developing dendrites, we co-labeled RGCs with GCaMP6s and mtDsRed. Simultaneous imaging of dendritic $Ca^{2+}$ signals and mitochondrial movements (*Figure 4E*), revealed that wave-associated $Ca^{2+}$ transients do not alter mitochondrial motility in RGC dendrites: neither the instantaneous motile fraction (*Figure 4H* and *Video 2*) nor the average speed of moving mitochondria ($\triangle$ speed: 0.132 ± 0.067 µm/s, n = 66 mitochondria, p>0.1) change following a dendritic $Ca^{2+}$ transient.

In cultured neurons, $Ca^{2+}$ elevations were shown to stop moving mitochondria via the adaptor protein Miro1, which uncouples mitochondria from kinesin motors upon $Ca^{2+}$ binding. Miro1 is readily detected in Western blots of the retina from at least P6 onwards and its expression reaches mature levels by P15 (*Figure 4—figure supplement 1*). In addition, we found that biolistic delivery of wild-type Miro1 or of a $Ca^{2+}$-binding-deficient mutant (Miro1KK) does not alter the motility of dendritic mitochondria. Thus, a lack of Miro1 expression or function is unlikely to account for the lack of effect of physiologic activity patterns on the movements of mitochondria in RGC dendrites.

Several previous studies raised external $K^+$ concentrations to elicit changes in mitochondrial motility. We therefore tested the effects of this manipulation in the developing retina. Elevating external $K^+$ to 30 mM tonically increased the concentration of $Ca^{2+}$ in RGC dendrites and, similar to observations in cultured neurons, decreased the motile fraction of mitochondria (*Figure 4F, I* and *Video 3*). By contrast, blocking cholinergic retinal waves and the accompanying $Ca^{2+}$ transients with the nicotinic antagonist DhβE did not affect mitochondrial motility in RGC dendrites (*Figure 4G, J* and *Video 3*).

A subset of RGCs (21 of 59 cells) exhibited local dendritic $Ca^{2+}$ transients at a similar frequency as global signals (*Figure 4—figure supplement 2*, local: 0.0102 ± 0.0055 events / s, n = 7 cells, global: 0.0141 ± 0.0022 events / s, n = 32 cells, p>0.4). Local transients have previously been observed in RGC dendrites in the developing chick retina, where they are caused by $Ca^{2+}$ release from internal stores (*Lohmann et al., 2002*). Like global signals, local $Ca^{2+}$ transients in developing mouse RGC dendrites do not appear to alter mitochondrial motility (*Figure 4—figure supplement 2*).

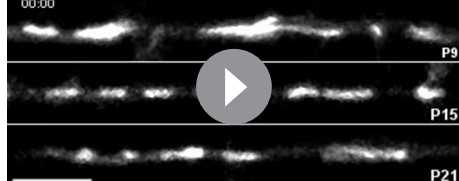

**Video 1.** Mitochondrial motility across development. Time-lapse confocal recording of RGC dendrites expressing mtYFP at P9 (*top panel*), P15 (*center panel*), and P21 (*bottom panel*). Scale bar: 5 µm. 0.9 Hz time-lapse, playback speed = 15 frames per second. mtYFP, mitochondrially targeted yellow fluorescent protein; RGC, retinal ganglion cell.

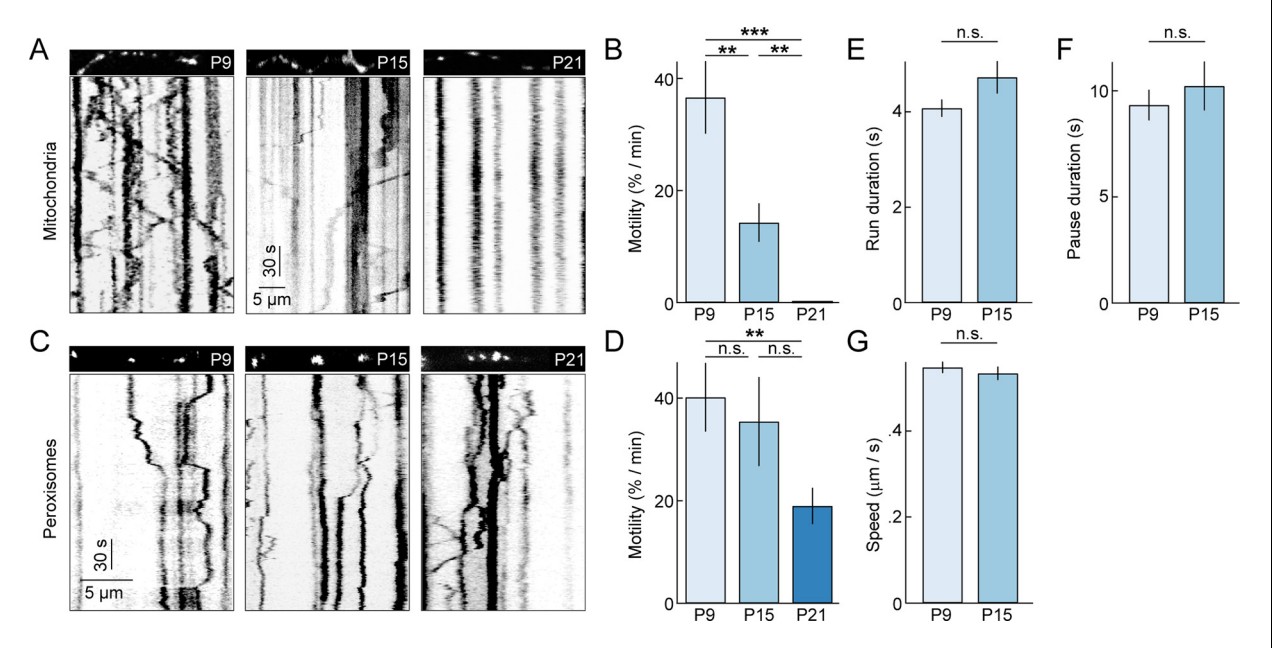

**Figure 3.** Motility of dendritic mitochondria and peroxisomes across development. (A) Kymographs of representative time-lapse imaging series of mitochondria (mtYFP, 0.9 fps, A) at P9 (*left*), P15 (*middle*), and P21 (*right*). *Top panels* show still frames at t = 0s of the branch segments depicted in the kymographs in the *bottom panels*. (B) Summary data of the motile fraction of mitochondria across development (P9 n = 10 RGCs, P15 n = 9 RGCs, P21 n = 7 RGCs). (C, D) Analogous to A (C) and B (D) but for time-lapse imaging of peroxisomes labeled with SKL-GFP (P9 n = 11 RGCs, P15 n = 10 RGCs, P21 n = 12 RGCs). (E–G) Bars (error bars) indicating the mean (± SEM) duration of uninterrupted runs (E), duration of pauses (F), and speed during uninterrupted motion (G) for mitochondria at P9 and P15 (P9 n = 65 mitochondria, P15 n = 32 mitochondria). See also **Video 1**. mtYFP, mitochondrially targeted yellow fluorescent protein; RGCs, retinal ganglion cells.

Together these results suggest that physiologically occurring activity patterns and $Ca^{2+}$ transients in the developing retina do not regulate mitochondrial motility in RGC dendrites, irrespective of Miro1 expression, and that tonic elevations of $Ca^{2+}$ upon application of high $K^+$ solution may engage different mechanisms to physiological transients.

## Sensory-evoked activity and dendritic mitochondria at maturity

As the retina matures, waves of spontaneous activity subside and light-evoked inputs begin to drive RGC activity. We wanted to test whether sensory-evoked activity affects the distribution and/or movements of mitochondria in RGC dendrites at maturity. Because mitochondrial movements are rare at P21, we acquired z-stacks of RGC dendrites in *Thy1-mtCFP-P* retinas every 15 min on a two-photon microscope. In the intervals between image acquisitions, retinas were either kept in darkness (*P21 – no stim*) or presented a full-field white noise stimulus with alternating contrast levels (*P21 – stim*, see 'Materials and methods'). MEA recordings confirmed that this stimulus robustly elevates the firing rates of RGCs (*Figure 5A–C*). In darkness, mitochondria rarely changed position even across two imaging intervals at P21, whereas approximately half of them were displaced over the same time at P9 (*Figure 5D, E*). Light stimulation did not alter the stability of mitochondria at P21 and very few changed their position in the imaging intervals (*Figure 5D, E*). Thus, similar to our observations for spontaneous activity in developing circuits, sensory-evoked activity does not appear to regulate the motility of dendritic mitochondria in the mature retina.

Toxins that depolarize mitochondria and inhibit their ability to synthetize ATP have been reported to increase, decrease or not change mitochondrial motility in axons and dendrites of different neuron types. To test how toxin-mediated stress affects mitochondrial transport in RGC dendrites, we applied a mixture of Antimycin A (a complex III inhibitor) and Oligomycin (an ATP synthase inhibitor) to P9 and P21 retinas. Fast (0.9 fps) and long-interval (15 min) time-lapse imaging revealed no

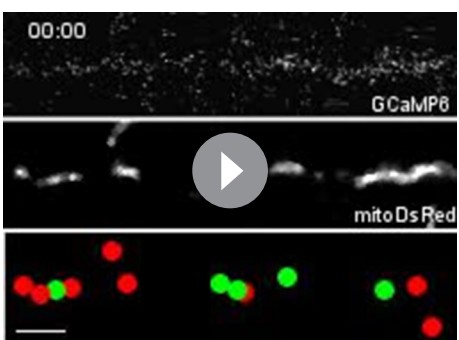

**Video 2.** Mitochondrial motility during spontaneous global Ca²⁺ transients. Simultaneous time-lapse two-photon recording of GCaMP6 and mtDsRed expression in P9 RGC dendrites. *Top panel* shows Ca²⁺ events in a section of dendrite. *Center panel* shows mtDsRed signal in the same dendritic branch. *Bottom panel* shows dots overlaid at each mitochondrion position tracked for mitochondria in the *center panel*. In frames in which a mitochondrion is moving, its dot is green; dots are red for stationary mitochondria. Scale bar: 5 µm. 0.9 Hz time-lapse, playback speed = 15 frames per second. RGC, retinal ganglion cell.

significant changes in the motility of mitochondria in developing or mature RGC dendrites (*Figure 5—figure supplement 1*).

## Pathological hyperactivity and dendritic mitochondria in retinal degeneration

Spontaneous oscillatory hyperactivity of RGCs is a common feature of retinal degenerations and originates in presynaptic circuits (*Borowska et al., 2011*; *Margolis et al., 2008*; *Soto and Kerschensteiner, 2015*; *Soto et al., 2012*; *Stasheff, 2008*; *Yee et al., 2012*). In *Crx⁻ᐟ⁻* mice, a model of Leber congenital amaurosis, retinal waves are preserved. But, from P15 on RGCs exhibit rhythmic hyperactivity as a result of enhanced input from BCs (*Soto et al., 2012*). We previously showed that this elevated activity prolongs and enhances BC-RGC synaptogenesis (*Soto et al., 2012*). Given the correlation between synaptic development and mitochondrial motility in wild-type mice (*Figures 2* and *3*), we first confirmed RGC activity was elevated in explants from P21 *Crx⁻ᐟ⁻* mice (*Figure 6A–C*), and then examined mitochondrial transport in RGC dendrites at this age. We found that dendritic mitochondria in *Crx⁻ᐟ⁻* retinas remained motile at P21, at a level intermediate to those at P9 and P15 in wild-type mice (*Figure 6D, E* and *Video 4*). Moreover, the movement patterns of motile mitochondria at P21 in *Crx⁻ᐟ⁻* mice except for slightly longer pauses were indistinguishable from those in wild-type at P9 and P15 (*Figure 6F–H*).

## Discussion

The importance of mitochondrial dynamics and distribution to neuronal function is highlighted by the fact that mutations that alter the fission-fusion balance and transport of mitochondria cause neurodegenerative diseases (*Chan, 2006*). Among the neurons affected by such diseases are RGCs, which degenerate in autosomal dominant optic atrophy (ADOA), the most common hereditary optic neuropathy (*Olichon et al., 2006*; *Yu-Wai-Man et al., 2011*). A majority of ADOA patients harbors mutations in the *OPA1* gene, which encodes a GTPase involved in mitochondrial fusion (*Olichon et al., 2006*; *Yu-Wai-Man et al., 2011*). Changes in fission-fusion balance alter the transport and distribution of mitochondria in neurons (*Chan, 2006*).

In axons, mitochondrial transport and distribution have been explored in detail, including recent in vivo imaging studies (*Breckwoldt et al., 2014*; *Misgeld et al., 2007*; *Plucinska et al., 2012*; *Takihara et al., 2015*), and the contributions of mitochondria to axonal branching and synaptic transmission are relatively well understood (*Courchet et al., 2013*; *Medler and Gleason, 2002*; *Spillane et al., 2013*; *Verstreken et al., 2005*; *Werth and Thayer, 1994*). By comparison, dendritic mitochondria have been studied less, and many aspects of their transport, distribution, and function, particularly in intact circuits, remain obscure.

We find that during development mitochondria enter RGC dendrites to near-mature levels before branching patterns are established (*Figure 2*). Similar observations were recently made for dendrites of cerebellar Purkinje cells (*Fukumitsu et al., 2015*), and together with experiments that block export of mitochondria into dendrites support the notion that local mitochondria are required for the formation and/or maintenance of dendritic branches (*Fukumitsu et al., 2015*; *Ishihara et al., 2009*). We find that during development, dendritic mitochondria gradually become enriched (>15-fold) near branch points (*Figure 2*). These mitochondria likely provide energy for lipid and protein biosynthesis, and secretory trafficking in the complex ER structures and Golgi outposts found at

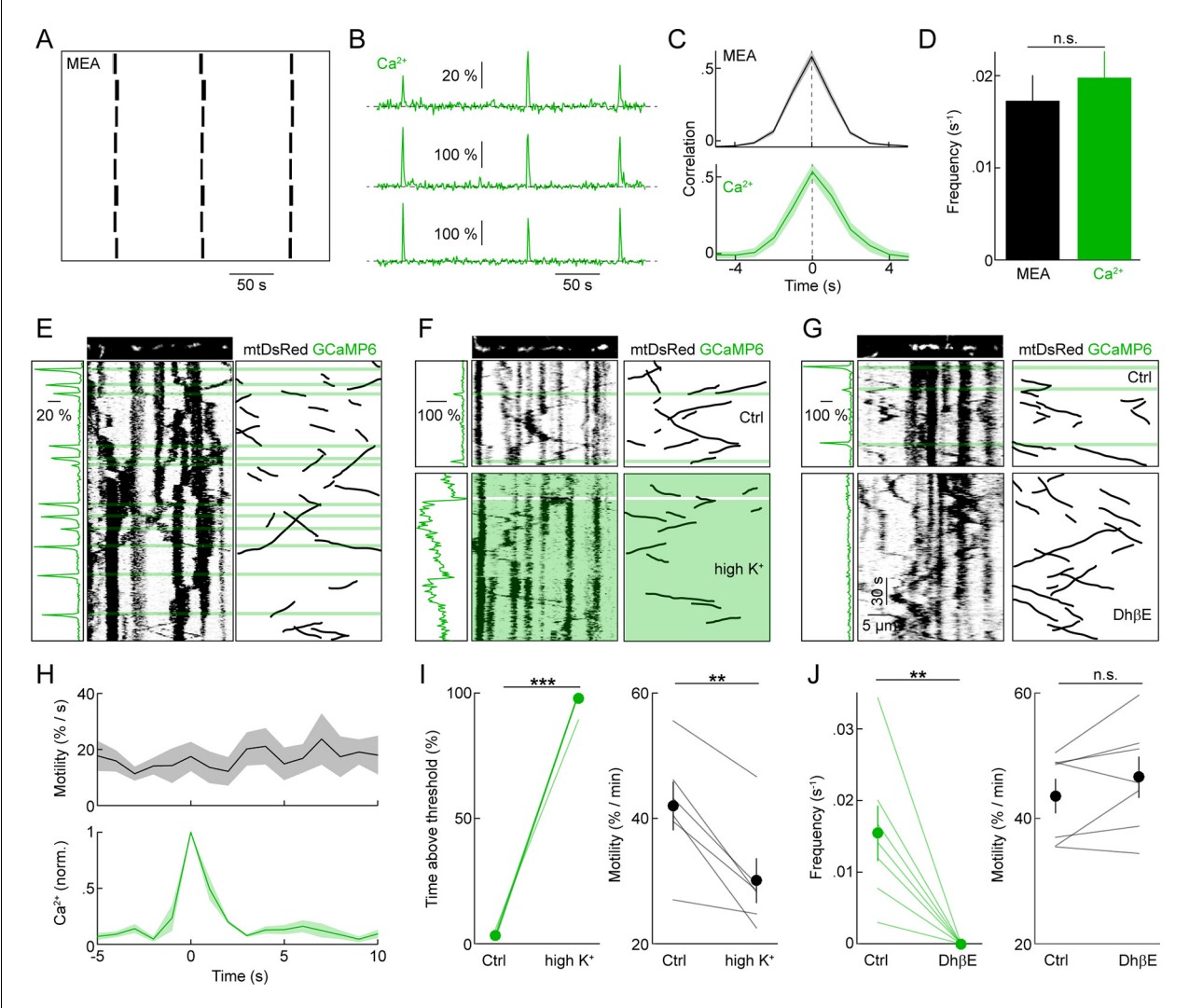

**Figure 4.** Spontaneous neuronal activity, dendritic Ca$^{2+}$ transients, and mitochondrial motility during development. (**A**) Raster plot of representative spike trains of eight neighboring RGCs recorded in a biolistically labeled retinal explant on an MEA at P9. (**B**) Representative △F/F traces of GCaMP6s signals recorded from three neighboring RGCs by two-photon imaging. (**C**) Cross-correlations of firing rates (*top panel*) and dendritic Ca$^{2+}$ transients (*bottom panel*) of neighboring RGCs (<200 μm between recording sites on MEA, <200 μm between cell bodies in Ca$^{2+}$ imaging) (MEA recordings n = 30 pairs, Ca$^{2+}$ imaging n = 11 pairs). (**D**) Average frequencies of waves of correlated activity on MEA (*black bar*) and global Ca$^{2+}$ signals in RGC dendrites observed by two-photon imaging (*green bar*) in biolistically labeled retinal explants at P9 (MEA n = 6 retinas, Ca$^{2+}$ imaging n = 22 retinas, p>0.6). (**E–G**) GCaMP6s and mtDsRed signals during simultaneous fast (0.9 fps) time-lapse two-photon imaging. *Top*: still frames of mtDsRed channel at t = 0s. *Left panels*: ΔF/F traces aligned with time plotted in kymograph. ΔF/F traces were calculated based on the average GCaMP6s intensity in the dendrite segment shown above the kymograph. *Center panels*: kymographs of mtDsRed signal. Green overlay indicates the timing Ca$^{2+}$ transients. *Right panels*: black lines represent schematized depictions of mitochondrial runs in the kymographs. (**F**) Break between *top* and *bottom panels* represents 10 min between pre-treatment (*top*) and 30 mM K$^{+}$ application (*bottom*). (**G**) 5 μM DHβE treatment. (**H**) *Top*: Plot of the instantaneous mitochondrial motility as a function of time relative to a Ca$^{2+}$ transient (n = 134 mitochondria, 10 RGCs). *Bottom*: Normalized Ca$^{2+}$ imaging trace as a function of time (aligned on t = 0 as the time of the Ca$^{2+}$ transient). *Lines (shaded areas)* represent the mean (± SEM). (**I, J**) Paired plots of RGCs before and after 30 mM K$^{+}$ (**I**) or DhβE (**J**) application. In the *left panel* of **I** the percentage of time above threshold measured was measured as percentage of frames in which the average GCaMP6s intensity within the frame was more than 2 SD above mean intensity of across the whole recording. The left panel of **J** show the frequency of Ca$^{2+}$ transients. *Right panels* of **I** and **J** show changes in mitochondrial motility between pre-treatment and treatment portions of the recordings (30 mM K$^{+}$ n = 8 RGCs, DhβE n = 7 RGCs). RGCs, retinal ganglion cells.

The following figure supplements are available for figure 4:

**Figure supplement 1.** Miro1 expression and effect on mitochondrial motility in RGCs during development.

*Figure 4 continued on next page*

*Figure 4 continued*

**Figure supplement 2.** Spontaneous, local dendritic Ca²⁺ transients and mitochondrial motility during development.

branch points (*Cui-Wang et al., 2012*; *Horton et al., 2005*; *Ye et al., 2007*). Consistent with this idea, ER structures and Golgi outposts at branch points are required for the formation and maintenance of dendritic branches (*Cui-Wang et al., 2012*; *Horton et al., 2005*; *Ye et al., 2007*), and the branching phenotypes of neurons lacking dendritic mitochondria can be rescued by exogenous supply of ATP-phosphocreatine (*Fukumitsu et al., 2015*; *Li et al., 2004*).

Mitochondria in RGC dendrites localize to excitatory synapses (*Figure 2*). Similar observations have been made on cultured neurons, although the extent of co-localization varied between studies (*Chang et al., 2006*; *Li et al., 2004*). Local mitochondria are thought to participate in the formation of synapses and their plasticity (*Ishihara et al., 2009*; *Li et al., 2010*; *Li et al., 2004*). Experiments in cultured hippocampal neurons suggested that enhancing the dendritic mitochondria content is sufficient to increase the number of synapses (*Li et al., 2004*). As mitochondrial density in RGCs dendrites reaches near-mature levels before most synapses are formed, it seems unlikely that mitochondria limit the rate of synaptogenesis during development of this circuit.

Mitochondrial transport in RGC dendrites is well described by a state diagram recently applied to axonal transport, in which mitochondria exist in a motile or a stationary state (*Obashi and Okabe, 2013*). Motile mitochondria alternate between short runs and short pauses (*Figure 3*) and frequently switch the direction of their movements (*data not shown*). By contrast, stationary mitochondria remain in place for long periods of time. Unlike transitions between runs and pauses in the motile state, transitions between motile and stationary state are rare (*Obashi and Okabe, 2013*). Our results show that as circuits mature, dendritic mitochondria undergo a nearly complete shift to the stationary state. Thus, whereas approximately 30% of mitochondria move at any time in P9 dendrites, we observed no mitochondrial movements at P21 during fast time-lapse imaging (0.9 fps, *Figure 3*) and displacements were rare even during long imaging intervals (15–30 min, *Figure 5*). These rare displacements may be associated with mitochondrial fission or support mitochondrial fusion or mitophagy (*Chen and Chan, 2009*; *Maday and Holzbaur, 2014*). Disruption of mitochondrial fusion in a mouse model of ADOA (*Opa1⁺/⁻* mice) results in shortened mitochondria, which accumulate in proximal RGC dendrites (*Williams et al., 2012*). As *Opa1⁺/⁻* mice age (>1 yr), the number of excitatory synapses and branches of RGC dendrites declines, highlighting the importance of mitochondrial

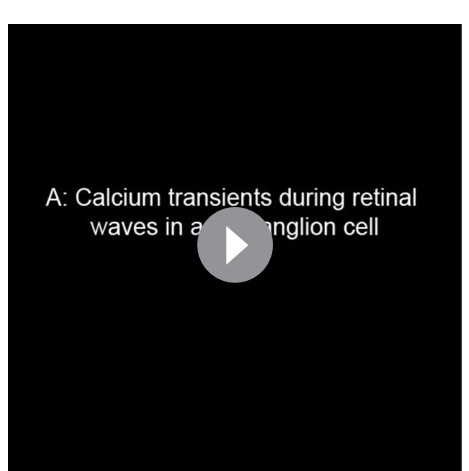

**Video 3.** Spontaneous global Ca²⁺ transients and pharmacological treatment during retinal waves. (A) Time-lapse two-photon recording of GCaMP6 signal in a P9 RGC undergoing spontaneous Ca²⁺ events while participating in retinal waves. (B) A P9 RGC before and during wash-in of 30 mM High K+ mACSF, exhibiting a large sustained increase in GCaMP6 intensity. (C) A P9 RGC before and during wash-in of 5 µM DhβE, exhibiting an abolition of spontaneous wave activity. Scale bars: 5 µm. 0.9 Hz time-lapse, playback speed = 20 frames per second. RGC, retinal ganglion cell.

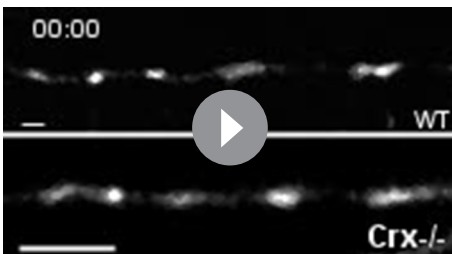

**Video 4.** Mitochondrial motility in pathologically hyperactive mature cells. Time-lapse confocal recording of RGC dendrites expressing mtYFP in a P21 WT dendrite (*top panel*) and a P21 Crx⁻/⁻ dendrite (bottom panel). Scale bars: 5 µm. 0.9 Hz time-lapse, playback speed = 15 frames per second.

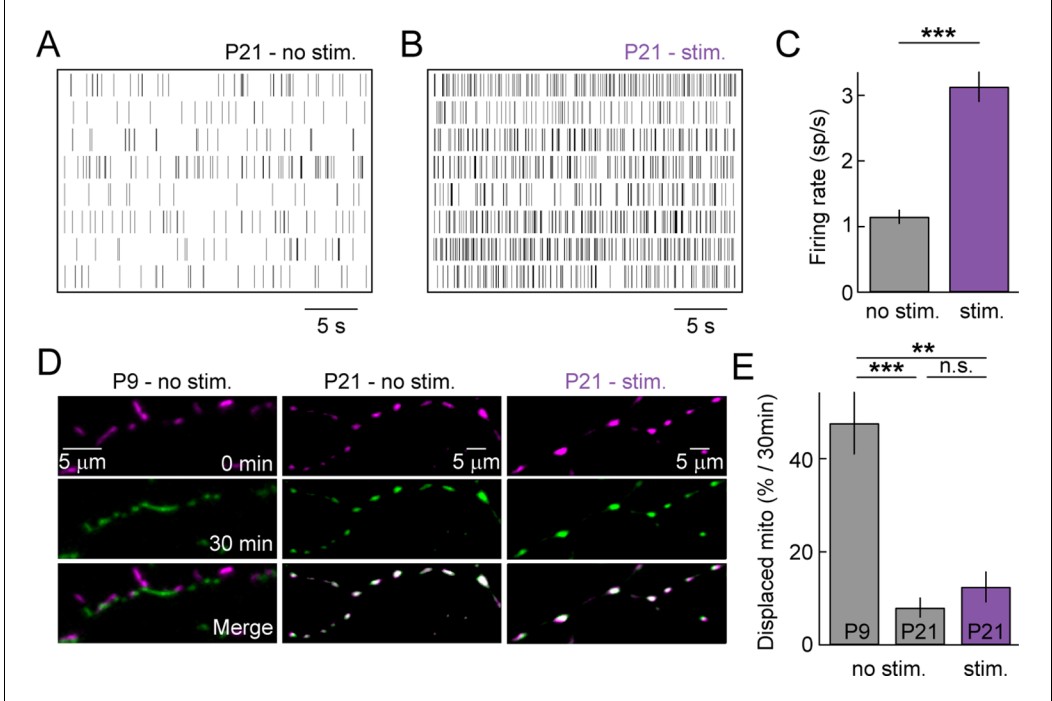

**Figure 5.** Sensory-evoked neuronal activity and mitochondrial motility at maturity. (**A**, **B**) Spike raster plots of eight representative RGCs recorded in darkness (**A**) and during presentation of a full-field white noise stimulus (**B**, see 'Materials and methods'). (**C**) *Bars (error bars)* indicate the mean (± SEM) firing rates of RGCs (n = 334 RGCs, 3 retinas, p<10$^{-26}$). (**D**) Representative RGCs expressing mtCFP at t = 0 min (*top panels*), t = 30 min (*middle panels*) after being exposed to white noise stimulus (*right panels*) or kept in darkness (*left and center panels*). *Bottom panels* show merged images of t = 0 and t = 30 min. (**E**) *Bars (error bars)* indicating the mean (± SEM) of % mitochondria displaced between t = 0 min and t = 30 min (P9 – no stim. n = 5 RGCs, P21 – no stim. n = 5 RGCs, P21 – stim n = 5 RGCs). RGCs, retinal ganglion cells.

The following figure supplement is available for figure 5:

**Figure supplement 1.** Antimycin A and Oligomycin and mitochondrial motility.

fusion for dendritic and synaptic integrity (*Williams et al., 2010*; *Williams et al., 2012*).

In isolated cortical neurons, a modest decline in mitochondrial motility was observed with increasing time in culture (*Chang and Reynolds, 2006*). In the intact retina, we discover a drastic developmental shift (motile to stationary) in dendritic mitochondria. Importantly, this shift is cargo-specific, as peroxisomes keep moving in dendrites of mature RGCs (*Figure 3*), and compartment-specific, as mitochondria remain motile in RGC axons (*data not shown*) (*Takihara et al., 2015*). Together with results on dendritic localization (*Figure 2*), these observations reveal that mitochondria form stable associations with synapses and branch points in RGC dendrites, and suggest that the prevailing view of mitochondria as highly motile organelles applies to developing but not to mature dendrites.

Using simultaneous two-photon imaging of Ca$^{2+}$ and mitochondria in the intact retina, we find that neither patterned spontaneous activity during development (*Figure 4*) nor sensory-evoked activity at maturity (*Figure 5*) regulate the motility of mitochondria in RGC dendrites. These results are in contrast to studies of cultured neurons, which suggest that neuronal activity controls motility of dendritic mitochondria via the adaptor protein Miro1 (*Li et al., 2004*; *MacAskill et al., 2009*; *Rintoul et al., 2003*). We confirmed that Miro1 is expressed in the developing and mature retina and showed that overexpression of wild-type Miro1 or a mutant unable to bind Ca$^{2+}$ (*Fransson et al., 2006*) do not affect mitochondrial motility in RGC dendrites. Thus, it seems unlikely that differences in Miro1 expression or function account for the discrepant results on the influence of activity. Instead, we suggest that differences between activity patterns physiologically occurring in intact circuits and artificial stimuli used to elicit effects on mitochondrial motility in culture may be a contributing factor. In support of this notion, we find that elevation of external K$^+$ reduces

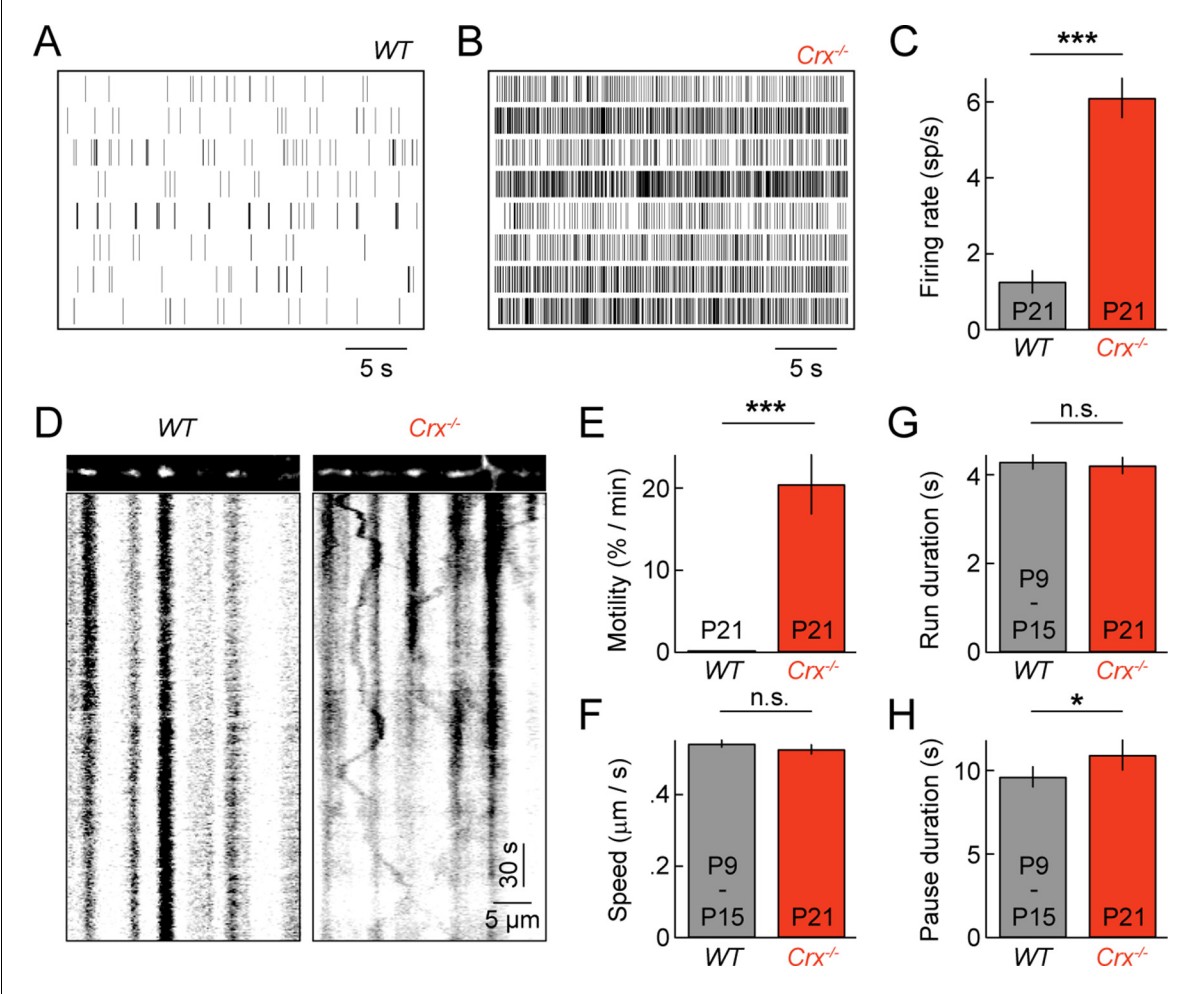

**Figure 6.** Pathological hyperactivity and dendritic mitochondria in retinal degeneration. (**A**, **B**) Spike raster plots of eight representative RGCs recorded from P21 *WT* (**A**) and from P21 *Crx*[-/-] retinas (**B**). (**C**) *Bars (error bars)* indicating the mean (± SEM) firing rates of *WT* and *Crx*[-/-] RGCs (*WT* n = 100 RGCs, 3 retinas, *Crx*[-/-] n = 235 RGCs, 3 retinas, p<10[-8]). (**D**) Kymographs of representative time-lapse series of a P21 *WT* RGC (*left panels*) and *Crx*[-/-] RGC (*right panels*). (**E**) *Bars (error bars)* indicating the mean (± SEM) motile fraction of mitochondria in P21 WT and *Crx*[-/-] dendrites (P21 *WT* n = 7 RGCs, *Crx*[-/-] n = 6 RGCs). (**FH**) *Bars (error bars)* indicating the mean (± SEM) mitochondrial speed during uninterrupted motion (**F**), duration of uninterrupted runs (**G**), and duration of pauses (**H**) for mitochondria in P9 – 15 *WT* and *Crx*[-/-] dendrites (*Crx*[-/-] n = 30 mitochondria, pooled *WT* P9 and P15 n = 97 mitochondria). RGC, retinal ganglion cell.

mitochondrial motility in RGC dendrites (*Figure 4*) as it does in cultured neurons (*Li et al., 2004*), but elicits tonic rises in intracellular $Ca^{2+}$ not observed during physiologic activity.

In a mouse models of retinal degeneration, circuits in the inner retina exhibit spontaneous hyper-activity (*Borowska et al., 2011*; *Margolis et al., 2008*; *Soto and Kerschensteiner, 2015*; *Soto et al., 2012*; *Stasheff, 2008*; *Yee et al., 2012*). In *Crx*[-/-] mice, we previously showed that this hyperactivity enhances and prolongs synaptogenesis between BCs and RGCs (*Soto et al., 2012*). Here, we find that extended synaptogenesis is matched by an increase in mitochondrial motility at P21 in *Crx*[-/-] mice. Together with the parallel decline in synapse turnover and mitochondrial motility across normal development (*Figure 3*) (*Kerschensteiner et al., 2009*; *Morgan et al., 2008*), this suggests that the two processes are linked.

In summary, we discover a cargo- and compartment-specific developmental shift in the transport of mitochondria, which during a critical period of high motility localize to synapses and branch points in RGC dendrites and subsequently maintain stable positions. In addition, using simultaneous two-photon imaging of $Ca^{2+}$ signals and mitochondrial transport, we find that neither spontaneous nor

sensory-evoked activity patterns regulate the motility of dendritic mitochondria in the intact retina. Together these results suggest important amendments to our understanding of dendritic mitochondria.

## Materials and methods

### Animals

All animals were handled according to a protocol (# 20140095) approved by the Animal Studies Committee of Washington University School of Medicine and performed in compliance with the National Institutes of Health Guide for the Care and Use of Laboratory Animals. *Thy1-mtCFP-P* (*Misgeld et al., 2007*) and *Crx⁻ᐟ⁻* (*Furukawa et al., 1999*) mice were backcrossed to C57BL/6J for more than five generations.

### Tissue preparation

Eyes were removed from mice deeply anesthetized with $CO_2$. Retinas were dissected from eye cups and prepared as flat mounts on permeable filter paper (Millipore, Sigma-Aldrich, Saint Louis, MO). Tissue was prepared in ice-cold mouse Artificial Cerebrospinal Fluid (mACSF) buffered with HEPES (biolistic experiments, concentrations in mM: 119 NaCl, 2.5 KCl, 2.5 $CaCl_2$, 1.3 $MgCl_2$, 1 $NaH_2PO_4$, 11 Glucose, and 20 HEPES - pH adjusted to 7.35–7.40 with NaOH) or sodium bicarbonate (light stimulation and MEA, concentrations in mM: 125 NaCl, 2.5 KCl, 1 $MgCl_2$, 1.25 $NaH_2PO_4$, 2 $CaCl_2$, 11 Glucose, and 26 $NaHCO_3$). Retinas used for biolistic transfection were incubated at 33°C in an $O_2$-perfused chamber overnight. Retinas used for light stimulus and MEA experiments were dissected under infrared illumination (excitation wavelength >900 nm) and incubated at 33°C in a light-tight chamber for approximately 2 hr before experiments. Live retinal tissue was constantly perfused at 1 ml/min with 33°C mACSF bubbled with $O_2$ (confocal experiments) or 95% $O_2$ / 5% $CO_2$ (two-photon experiments) during light stimulation and imaging. In pharmacologic experiments, 5 µM DhβE (Tocris Bioscience) or 4 µM Antimycin A (Enzo Life Sciences) and 10 µM Oligomycin (Sigma-Aldrich) were added to mACSF. To raise extracellular $K^+$ concentrations to 30 mM, KCl was substituted for NaCl in mACSF (concentrations in mM: 97.5 NaCl, 30 KCl, 1 $MgCl_2$, 1.25 $NaH_2PO_4$, 2 $CaCl_2$, 11 Glucose, and 26 $NaHCO_3$).

### Biolistic transfection

A gene gun was used to transfect RGCs in the ex vivo retinal flat-mount preparations as described previously (*Kerschensteiner et al., 2009*; *Morgan et al., 2008*). Briefly, plasmids encoding fluorescent proteins were precipitated onto 1.6 µm gold particles (Bio-Rad) and gold particles loaded into Tefzel tubing (Bio-Rad) (*Morgan and Kerschensteiner, 2012*). Helium pressure at 40 psi was used to deliver gold particles to flat-mounted retinal tissue (*Morgan and Kerschensteiner, 2011*).

### MEA recordings

Multi-electrode array (MEA) recordings of RGC action potentials were acquired as described previously (*Pearson and Kerschensteiner, 2015*). Briefly, rectangular pieces of isolated dorsal retinal tissue were mounted onto planar arrays of 252 electrodes (MultiChannelSystems). Spike waveforms recorded at each electrode were used to sort activity into trains of RGC action potentials using principal component analysis (Offline Sorter; Plexon). Correlation coefficients normalized for nonstationary firing rates were calculated as described previously (*Kerschensteiner and Wong, 2008*).

### Image acquisition and analysis

Fixed and live retinas were imaged on Olympus Fv1000 laser scanning confocal and two-photon microscopes using 60x 1.35 NA oil-immersion (fixed retinas), 60x 1.1 NA water-immersion, and 20x 0.95 NA water-immersion objectives (live retinas). Images were processed and analyzed using ImageJ (NIH), Amira (FEI), and software written in MATLAB (MathWorks).

To analyze mitochondrial and synaptic distributions, retinas were fixed in 4% paraformaldehyde for 30 min at RT. Synapses were identified and dendrites skeletonized using custom MATLAB software described previously (*Kerschensteiner et al., 2009*; *Morgan et al., 2008*). Briefly, the cytosolic tdTomato signal was used to create binary masks (Amira) of RGC dendrites. Dendrites were then

skeletonized into 1 µm-long linked segments based on these masks. Synapses were identified by iterative thresholding of PSD95-CFP signal within the dendritic mask (*Kerschensteiner et al., 2009*; *Morgan et al., 2008*). Voxels belonging to mitochondria were similarly identified by local thresholding algorithms on mtYFP signal within the dendritic mask. The density of mitochondria in dendrites was expressed as a volumetric fraction based on the number of voxels assigned to mitochondria and that contained in the dendritic mask. To determine quantitatively whether mitochondria localize to synapses, the observed average nearest neighbor distance (synapse to mitochondrion) for a given RGC was compared to the distribution of average nearest distances obtained in Monte Carlo simulations in which the position synapses of the same cell was randomized along its dendrites. For all cells, the mitochondria were closer to real synapses than their simulated counterparts.

Time-lapse recordings of mitochondrial and peroxisomal motility, and $Ca^{2+}$ waves were acquired at 0.9 frames per second (fps). Motile fractions were calculated as (motile organelles)/(motile organelles + stationary organelles). We refer to the motile fraction as the fraction moving during 1 min of observation, whereas instantaneous motile fraction is used to denote the fraction moving between consecutive frames in 0.9 fps imaging series. Kymographs were generated using the ImageJ Multiple Kymograph plugin. Mitochondrial speeds and peroxisomal speeds and mitochondrial run and pause durations were calculated using custom MATLAB software and organelle positions tracked manually using the ImageJ MTrack2 plugin.

Mitochondrial displacement during light stimulation of P21 retinas was computed using custom MATLAB software. mtCFP signal was masked in Amira to include only mitochondria in contiguous dendritic branches. mtCFP intensity was binarized and MATLAB software was used to identify regions of connected pixels as individual mitochondria. Images at t = 30 min were subtracted from images at t = 0 min; mitochondria identified at 0 min that did not have at least 20% overlap with a mitochondrion at 30 min was counted as a displaced mitochondrion. Percent displacement was calculated as 100 x (displaced mitochondria)/(total mitochondria). Stimulated retinas were shown light stimulus patterns of alternating high- and low-contrast (mean intensity 5000 R*/rod/s) periods lasting 60 s each for a total of 15 min. Image stacks of mtCFP signal throughout RGC dendrites were collected between light stimulations. Unstimulated retinas were kept in darkness for 15 min between image stacks.

## Acknowledgements

We thank Dr. M Kerschensteiner for comments on the manuscript and are grateful to members of the D Kerschensteiner lab for helpful discussions. This work was supported by grants from the National Institutes of Health (EY021855 to DK and EY013360 to MF).

## Additional information

### Funding

| Funder | Grant reference number | Author |
|---|---|---|
| National Institutes of Health | EY021855 | Daniel Kerschensteiner |
| National Institutes of Health | EY013360 | Michelle C Faits |

The funders had no role in study design, data collection and interpretation, or the decision to submit the work for publication.

### Author contributions

MCF, DK, Conception and design, Acquisition of data, Analysis and interpretation of data, Drafting or revising the article; CZ, Acquisition of data, Analysis and interpretation of data; FS, Conception and design, Acquisition of data, Analysis and interpretation of data

### Ethics

Animal experimentation: All animals were handled according to protocols approved by the Animal Studies Committee of Washington University School of Medicine (Protocol#: 20140095) and

experiments were performed in compliance with the National Institutes of Health Guide for the Care and Use of Laboratory Animals.

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
