## [Decision Letter]

Thank you for submitting your work entitled "Dendritic mitochondria reach stable positions during circuit development" for consideration by *eLife*. Your article has been favorably evaluated by Richard Aldrich (Senior editor) and three reviewers, one of whom is a member of our Board of Reviewing.

The reviewers have discussed the reviews with one another and the Reviewing editor has drafted this decision to help you prepare a revised submission.

Summary:

The authors quantify mitochondrial location and trafficking rates in dendrites during retinal ganglion cell development in situ and find that trafficking stops upon neuron maturation. The timing of localization of mitochondria to synapses and dendrite branch points differs during development. The authors also find that calcium waves reported previously to halt mitochondrial trafficking via Miro1 do not do so under more physiological conditions. Mitochondrial motility was retained during development in a model of retinal degeneration (Leber congenital amaurosis), where synaptogenesis between bipolar and retinal ganglion cells is elevated and prolonged. The conclusions of these well controlled experiments performed under fairly physiological conditions contrast with those from several prior papers using isolated cultured neurons and would have substantial impact on the field.

Essential revisions:

1) Mutations in Opa1 leading to DOA indicate that mitochondrial fusion is essential for retinal ganglion cells, likely involving dendrites (see point 2 below), possibly to mitigate mitochondrial damage or to support mitophagy by balancing mitochondrial fission. As mitochondrial fusion would demand some degree of motility, assessing mitochondrial trafficking after applying mitochondrial stress would be important. Assessing effects of mitochondrial toxins such as Rotenone, Oligomycin or Antimycin A applied to the retinal cultures at P9 and P21 and determining if and in what way the stressors alter movement may yield insight into the apparent lack of motility despite the essential role of mitochondrial fusion. Perhaps the motility involved in fusion is on a much slower time scale than the motility involved in trafficking at P9. Some mention of the essential role of Opa1 particularly in retinal ganglion cells and how this mitochondrial fusion requirement relates to the level of motility studied here would strengthen the Discussion.

2) Furthermore, the two papers below that support the necessity of mitochondrial dynamics to maintain dendritic mitochondria should be cited and discussed. The first pathological changes in a mouse model of DOA with mutant OPA1 (defective fusion) actually affect dendrites before impairing axons and cell bodies in retinal ganglion cells (Williams PA, et al. Opa1 deficiency in a mouse model of dominant optic atrophy leads to retinal ganglion cell dendropathy. Brain 2010;133(10):2942-51 and Williams PA, et al. Opa1 is essential for retinal ganglion cell synaptic architecture and connectivity. Brain 2012;135:493-505).

3) Do calcium transients increase along with the retinal wave increase in RGCs from *Crx^-/-^* mice? It would be important to assess the extent of calcium influx change to test/corroborate the important conclusion of the authors that physiological levels of calcium transients do not halt mitochondrial transport.

---

## [Author Response]

*1) Mutations in Opa1 leading to DOA indicate that mitochondrial fusion is essential for retinal ganglion cells, likely involving dendrites (see point 2 below), possibly to mitigate mitochondrial damage or to support mitophagy by balancing mitochondrial fission. As mitochondrial fusion would demand some degree of motility, assessing mitochondrial trafficking after applying mitochondrial stress would be important. Assessing effects of mitochondrial toxins such as Rotenone, Oligomycin or Antimycin A applied to the retinal cultures at P9 and P21 and determining if and in what way the stressors alter movement may yield insight into the apparent lack of motility despite the essential role of mitochondrial fusion. Perhaps the motility involved in fusion is on a much slower time scale than the motility involved in trafficking at P9. Some mention of the essential role of Opa1 particularly in retinal ganglion cells and how this mitochondrial fusion requirement relates to the level of motility studied here would strengthen the Discussion.*

As suggested, we performed additional live imaging experiments to test the effects of Antimycin A (a complex III inhibitor) and Oligomycin (an ATP synthase inhibitor) on mitochondrial motility in RGC dendrites. Toxin-mediated stress had previously been reported to have varied effects on mitochondrial transport, possibly depending on the neuron type and compartment (axon vs. dendrite) examined. Both at P9 and at P21 we found that Antimycin A and Oligomycin did not affect the motility of dendritic mitochondria in RGCs. These data are presented in a supplementary figure (Figure 5—figure supplement 1) in our revised manuscript and described in the Results section. In addition, we added a paragraph on the effects of Opa1 deficiency on RGC dendrites and their synaptic connections to the Discussion section of our revised manuscript.

*2) Furthermore, the two papers below that support the necessity of mitochondrial dynamics to maintain dendritic mitochondria should be cited and discussed. The first pathological changes in a mouse model of DOA with mutant OPA1 (defective fusion) actually affect dendrites before impairing axons and cell bodies in retinal ganglion cells (Williams PA, et al. Opa1 deficiency in a mouse model of dominant optic atrophy leads to retinal ganglion cell dendropathy. Brain 2010;133(10):2942-51 and Williams PA, et al. Opa1 is essential for retinal ganglion cell synaptic architecture and connectivity. Brain 2012;135:493-505).*

In the Discussion section of our revised manuscript, we speculate that the rare mitochondrial displacements we observe in long-interval time-lapse imaging experiments at P21 may be associated with mitochondrial fission, or support mitochondrial fusion or mitophagy. In this context we also discuss the evidence obtained in *Opa1_+/-_*mice, which highlights the importance of mitochondrial fusion to dendritic and synaptic integrity, and cite the studies mentioned by the reviewers.

*3) Do calcium transients increase along with the retinal wave increase in RGCs from Crx^-/-^ mice? It would be important to assess the extent of calcium influx change to test/corroborate the important conclusion of the authors that physiological levels of calcium transients do not halt mitochondrial transport.*

Spontaneous activity in *Crx_-/-_*mice is greatly increased with RGCs firing sustained at approximately 6 Hz. These firing rates are too high to distinguish individual firing events using GCaMP6s (s. Chen et al. Nature 2014). We tried to resolve firing events using a faster Ca^2+^ indicator: OGB-1. We simultaneously recorded spike trains (via patch clamp) and imaged dendritic Ca^2+^ signals (via 2-photon imaging) from individual RGCs in *Crx_-/-_*mice. As shown in the figure below (Figure 7), however, these experiments showed that even with OGB-1 firing rates are too high to resolve individual events by Ca^2+^ imaging. Dendritic Ca^2+^ levels were reduced when hyperpolarizing currents were injected to suppress spiking (data not shown). We chose not to include these data in our revised manuscript as similar experiments have been performed in a variety of systems. Our results therefore amount to a confirmation of already described kinetics of Ca^2+^ indicators, and their inclusion, we feel, would distract from the novel findings of our study. We hope that the reviewers agree with this choice.

Author response image 1.(**A**) 2-photon image of a section of RGC dendrite filled with OGB-1 via a recording electrode.The green line indicates the position and extent of the line scan. (**B**) △F/F of the OGB-1 signal (top, green) aligned with the voltage trace (bottom, black) recorded simultaneously from the same RGC in a *Crx-/-* retina.**DOI:**
http://dx.doi.org/10.7554/eLife.11583.016